# Optimization of chromium (VI) reduction in aqueous solution using magnetic Fe₃O₄ sludge resulting from electrocoagulation process

Pınar Belibagli[1], Zelal Isik[2], Nadir Dizge[2]*, Deepanraj Balakrishnan[3], Abdul Rahman Afzal[4], Muhammad Akram[5]*

1 Department of Energy Systems Engineering, Tarsus University, Tarsus, Turkey, 2 Department of Environmental Engineering, Mersin University, Mersin, Turkey, 3 Department of Mechanical Engineering, College of Engineering, Prince Mohammad Bin Fahd University, Al-Khobar, Saudi Arabia, 4 Department of Industrial Engineering, University of Business and Technology (UBT) Dahban, North Jeddah, Saudi Arabia, 5 School of Design, Informatics and Business, Abertay University, Dundee, United Kingdom

* m.akram@abertay.ac.uk (MA); ndizge@mersin.edu.tr (ND)

**Editor:** Santhana Krishna Kumar Alagarsamy, AGH Faculty of Mining Surveying and Environmental Engineering: Akademia Gorniczo-Hutnicza im Stanislawa Staszica w Krakowie Wydzial Geodezji Gorniczej i Inzynierii Srodowiska, POLAND

## Abstract

The reuse of electro-coagulated sludge as an adsorbent for Cr(VI) ion reduction was investigated in this study. Electro-coagulated sludge was obtained during the removal of citric acid wastewater by the electrocoagulation process. The following parameters were optimized for Cr(VI) reduction: pH (5–7), initial Cr(VI) concentration (10–50 mg/L), contact time (10–45 min), and adsorbent dosage (0.5–1.5 g/L). Cr(VI) reduction optimization reduction experimental sets were designed using response surface design. Cr(VI) reduction optimization results 97.0% removal efficiency and 15.1 mg/g adsorption capacity were obtained at pH 5.0, 1.5 g/L electro-coagulated Fe₃O₄ sludge, 10 mg/L initial Cr(VI) concentration and 45 min reaction time. According to the isotherm results, the experimental data are compatible with the Freundlich isotherm model, and since it is defined by the pseudo-second order model emphasizes that the driving forces of the Cr(VI) reduction process are rapid transfer of Cr(VI) to the adsorbent surface. The reusability of the adsorbent was investigated and Cr(VI) reduction was achieved at a high rate even in the 5th cycle. All these results clearly show that electro-coagulated Fe₃O₄ sludge is an effective, inexpensive adsorbent for Cr(VI) removal from wastewater.

## 1. Introduction

The global scarcity of freshwater resources is increasing along with rapid economic development and environmental hazards such as water pollution. At the same time, it becomes even more critical when combined with the insufficiency of water resources. Therefore, it has become an urgent necessity to quickly solve the water pollution problem. It is becoming increasingly important to address the global water pollution issue and water resource scarcity

**Data Availability Statement:** All relevant data are within the manuscript and its Supporting Information files.

**Funding:** The author(s) received no specific funding for this work.

**Competing interests:** The authors have declared that no competing interests exist.

[1,2]. Disadvantages such as metal pollution in aquatic environments, toxicity, environmental persistence and potential to pass into the food chain constitute an alarming situation. Heavy metals stand out as a primary contributor to water pollution on a global scale, particularly in groundwater. These pollutants infiltrate water systems from various origins, including synthetic fertilizers and chemicals employed in agricultural regions, alongside untreated industrial wastewater and excessive water extraction practices. Consequently, this poses a severe threat to water resource quality and imposes detrimental effects on ecosystems [3]. This may accumulate in aquatic habitats, microorganisms, aquatic flora, and fauna and negatively affect human health in the long term [1,2].

The use of heavy metals such as chromium (Cr), cadmium (Cd), and lead (Pb) is increasing in various industries, especially in manufacturing processes [4]. Wastewater from these manufacturing industries has high concentrations of Cr(VI) ranging from 5.0 to 280,000 mg/L in water as soluble ions such as hydrogen chromate ($HCrO_4^-$), dichromate ($Cr_2O_7^{2-}$), and chromate ($CrO_4^{2-}$) [5]. High chromium (Cr) concentrations in wastewater have an important place as a problem where water pollution can cause negative effects on human health. In order to control this problem, various physical and physicochemical traditional methods are used to regulate environmental impacts in wastewater treatment plants [6]. Adsorption is regarded as an excellant technique due to its higher efficiency, low implementation costs, ample accessibility, and ease of design in comparison to other methods [4]. One popular technique for removing impurities from aqueous solutions is adsorption. However, selecting cost-effective, eco-friendly, and highly efficient adsorbents for the chromium (Cr) removal from wastewater remains a significant challenge. To overcome this challenge, efforts are ongoing to develop safe and efficient adsorbents to effectively remove chromium from wastewater [3,7]. Traditional adsorbents used for the treatment of wastewater have disadvantages such as requiring additional separation to be removed from the solution, low reusability, and causing secondary pollution in water [8]. Adsorbents endowed with magnetism are not only cost-effective and environmentally friendly, but their robust magnetic characteristics also facilitate effortless recovery post-adsorption, rendering them highly attractive as a promising alternative [9].

Many successful research studies were conducted on Cr(VI) removal using adsorbents with magnetic properties [10–20]. However, the magnetic adsorbents used were synthesized by chemical synthesis. Recently, a few studies have eliminated various pollution in water by using aluminum or iron-containing sludges formed as a result of the electrocoagulation process (EC). Electro-coagulation (EC) is a process that produces minimal volumes of sludge primarily composed of metallic oxide/hydroxide. This EC sludge exhibits characteristics such as ease of dewatering, quick settling, resistance to acidity, stability, and convenient obtainability through filtration. Given these distinct benefits, leveraging electro-coagulated sludge for wastewater adsorption shows promise in terms of efficient sludge management and resource reutilization [21,22].

The generation of sludge in an EC (Electrocoagulation) process is deemed as waste, leading to escalated operating expenses due to waste disposal requirements. Briefly, the content of the sludge formed after the EC process consists of metal hydroxides (MH) and metal oxyhydroxides (MOH) types, depending on the electrode material used [22]. The primary interphase byproducts of electrochemically induced wastewater treatment with iron/steel electrodes are mainly iron hydroxide $Fe(OH)_2$ and ferric hydroxide Fe(OH). Crystal phases resembling those of magnetite have been observed in the sludge that results from the EC process with Fe electrodes [21,22].

Aragaw et al., [21] studied the adsorbent potential of raw and calcined iron hydroxide/oxide sludge after EC for direct red 28 (DR28) dye removal. Various parameters, including dye concentration, dosage, pH and temperature were optimized using both raw EC sludge and

calcined EC sludge. As a result of the data obtained, 1.262 mg/g and 1.252 mg/g adsorption capacity of raw and calcined sludges, respectively, were obtained for DR 28 dye removal [21]. García-Gómez et al. [23] investigated the ability of electrocoagulated aluminium (Al) sludge, produced through an electrocoagulation method using aluminium electrodes, to remove fluoride ions and arsenic. The results indicated that electrocoagulated Al sludge could serve as an effective adsorbent option [23]. Yamba et al. [24] investigated the sulfate removal potential of iron-containing electrocoagulated sludge obtained by using iron sheets in the EC system. The removal efficiency for both synthetic water and real water was found to be 99.1% and 98.7%, respectively [24]. Amri et al. [25] conducted a study to examine the removal potential of acid red 18 (AR 18) from water was investigated with Al-containing electrocoagulated sludge generated from Al electrodes. As a result of the optimization, AR18 dye adsorption capacity qmax was reported as 51.62 mg/g using Al-containing electrocoagulated adsorbents [25]. Apart from the above research, EC sludge is also used as fertilizer, pigment, construction material, and catalyst. However, although EC sludge has already been found to be valorizable, further studies are needed on the evaluation of EC sludge and the quality of sludge produced from the wastewater of EC processes [22].

Consequently, the EC process yields sludge that is deemed as waste, leading to heightened operational expenses associated with waste disposal. In particular, the use of electro-coagulated sludges formed after EC has a significant place in terms of reducing or reusing valuable waste production based on circular economy principles or current legislation. At the same time, as far as we know in the literature, the use of iron-containing sludge for Cr(VI) removal is the first contribution. For this reason, the sludge obtained after EC makes a significant contribution to the literature by removing Cr(VI), which is thought to be dangerous for humans and ecology. In this study, iron-containing sludge, which has magnetism properties resulting from citric acid removal using Fe electrodes, was used as an adsorbent and its Cr(VI) reduction potential was investigated. In batch Cr(VI) reduction experiments, parameters such as pH, dose, initial Cr(VI) concentration and temperature were optimized. SEM-EDS, XRD and zeta potential analyzes were performed for the characterization of electro-coagulated sludge. ANOVA statistical analysis, isotherm, kinetic and thermodynamic calculations were made with the data obtained as a result of the experiment.

## 2. Materials and methods

### 2.1. Cr(VI) reduction optimization employing statistical analysis and the Box-Behnken method

Response Surface Method (RSM) is an advanced experimental technique used for response optimization. Box-Behnken method (BBD), RSM is used to determine the levels of different factors affecting the response to be optimized. This technique can estimate first- and second-order factors. The designs are based on three-level incomplete factorial designs and include rotatable quadratic patterns [26]. In this study, the sludge obtained as a result of the optimization of the electrocoagulation process using iron electrodes of citric acid, which was carried out in the previous study [26], was used. The optimum conditions for the EC process are 20 V, pH 2.0, and a reaction time of 120 min. The sludge used is called electrocoagulated $Fe_3O_4$ sludge.

Before designing the reduction experiments, preliminary experiments were carried out and reduction parameters were determined. pH, adsorbent dose, Cr(VI) concentration, and time were selected as parameters. After preliminary experiments, the experimental design was planned at three different levels: 1, 0 and +1. Table 1 shows the range of variables selected for the reduction process. At the same time, it was determined whether the designed model was

**Table 1. All variable factors for Cr(VI) reduction.**

| Variable | Unit | Factor | Low | High |
|---|---|---|---|---|
| pH | - | A | 5 | 7 |
| Adsorbent Dose | g/L | B | 0.5 | 1.5 |
| Cr(VI) Concentration | mg/L | C | 10 | 50 |
| Reaction Time | min | D | 15 | 45 |

compatible or not by using ANOVA statistical analysis. The model designed offers a mathematical connection between the variables and the experimental data, which can be represented by a second-order polynomial equation (Eq (1)). [27].

$$Response = \beta_0 + \beta_1 A + \beta_2 B + \beta_3 C + \beta_5 AB + \beta_6 AC + \beta_7 BC + \beta_8 A^2 + \beta_9 B^2 + \beta_{10} C^2 \quad (1)$$

In this equation, the numbers 1, 2, 3,. . ., 14 represent the coefficients of the regression equation.

For reduction experiments, a stock solution of Cr(VI) was initially prepared. The concentrations required for the reduction studies were then derived by dilution from this prepared stock solution (Fig 1). Experiments were performed in duplicate and graphs were created by taking their averages. The Cr(VI) reduction capacity and reduction efficiency were determined using Eqs (2) and (3).

$$R(\%) = \frac{(C_i - C_e)}{C_i} * 100\% \quad (2)$$

$$q_e = \frac{(C_i - C_e) * V}{m} \quad (3)$$

Here, V represents the volume (liters), Ci stands for the initial Cr(VI) concentration (mg/L), Ce denotes the final Cr(VI) concentration (mg/L), and m indicates the mass of the electro-coagulated $Fe_3O_4$ sludge (gram).

## 2.2. Kinetic model and analysis of reduction isotherms for Cr(VI) reduction

In this study, an isotherm study was conducted to explain the interaction between adsorbent and adsorbate. The isotherm equations of the experiments carried out in Cr(VI) reduction

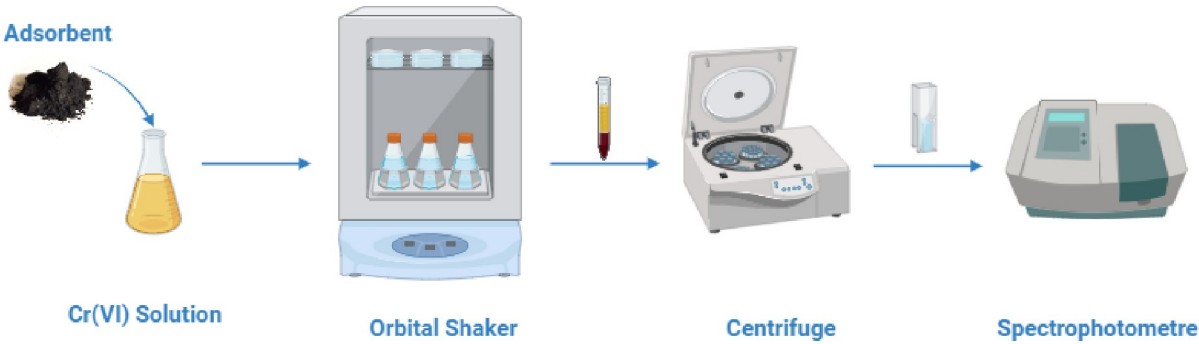

**Fig 1. General scheme of Cr(VI) reduction experiments.**

with electro-coagulated $Fe_3O_4$ sludge are given in S1 Table. For kinetics studies, the pseudo-first-order model (PFO) proposed by Lagergren [28], the pseudo-second-order (PSO) model, and the intra-particle diffusion (ID) model developed by Ho et al. in 1996 were employed. Detailed description of the kinetic models is given in S2 Table.

## 2.3. Reduction thermodynamic model of Cr(VI) reduction

Thermodynamic studies were carried out at different temperatures (30–40°C) under optimum conditions. Gibbs free energy (ΔG), total enthalpy changes (ΔH), and entropy changes (ΔS) square measure the physical science parameters [29]. Reduction thermodynamic experiment calculated with Eqs 4–6.

$$\Delta G = -RTLnK_{eq} \tag{4}$$

$$\Delta G° = \Delta H° - T\Delta S° \tag{5}$$

$$LnK_L = -\frac{\Delta H}{RT} + \frac{\Delta S}{R} \tag{6}$$

## 2.4. Fe$_3$O$_4$ electro-coagulated sludge characterization and analytical analysis

SEM, EDX and XRD analyzes were performed for structural and elemental analysis of electro-coagulated $Fe_3O_4$ sludge. Cr(VI) concentrations were analyzed using the 1.5 DPC method described by Alterkaoui et al [29].

## 3. Results and discussion

### 3.1. Characterization of iron-including electro-coagulated sludge

Fig 2A–2D shows SEM images of ferrous electro-coagulated sludge before and after Cr(VI) reduction at different magnifications. When the Fig 2A and 2B is examined, it can be seen that the morphological feature of the electro-coagulated sludge is that $Fe_3O_4$ powders with a rough, highly heterogeneous and agglomerated structure are formed [30,31]. Fig 2C and 2D shows SEM images after Cr(VI) reduction. When Fig 2C and 2D is examined, it is clearly seen that the structure of the iron-containing electrocoagulation sludge differs after Cr(VI) reduction. Kumar et al., [32] reported the appearance of shiny particles on the adsorbent surface and stated that there was a change in the structure after Cr(VI) reduction.

Fig 3A shows the EDS spectrum results of $Fe_3O_4$. According to EDS spectrum analysis, the dominant elements in the electro-coagulated sludge are 79.65% Fe, 12.85% O and 7.5% C. The 7.5% C element in the structure of electro-coagulated sludge is due to the removal of citric acid ($C_6H_8O_7$) in electrocoagulation experiments [26]. The graph of EDS after Cr(VI) reduction is shown in Fig 3B. After Cr(VI) reduction, 6.84% C, 30.19% O, 62.57% Fe and 0.40% Cr elements are dominant and these results show that Cr(VI) reduction has occurred.

In Fig 4A, the surface charge of the electro-coagulated $Fe_3O_4$ sludge was measured by zeta potential analysis at different values of pH from 2.00 to 7.00. The surface charge of electrocoagulated $Fe_3O_4$ sludge up to pH 3.00 was in the positive region. Beyond pH 3, the surface charges of WWP were in the negative region. When the solution pH increased from 2.00 to 7.00, the surface charge of WWP decreased from +4.24 mV to −30.43 mV.

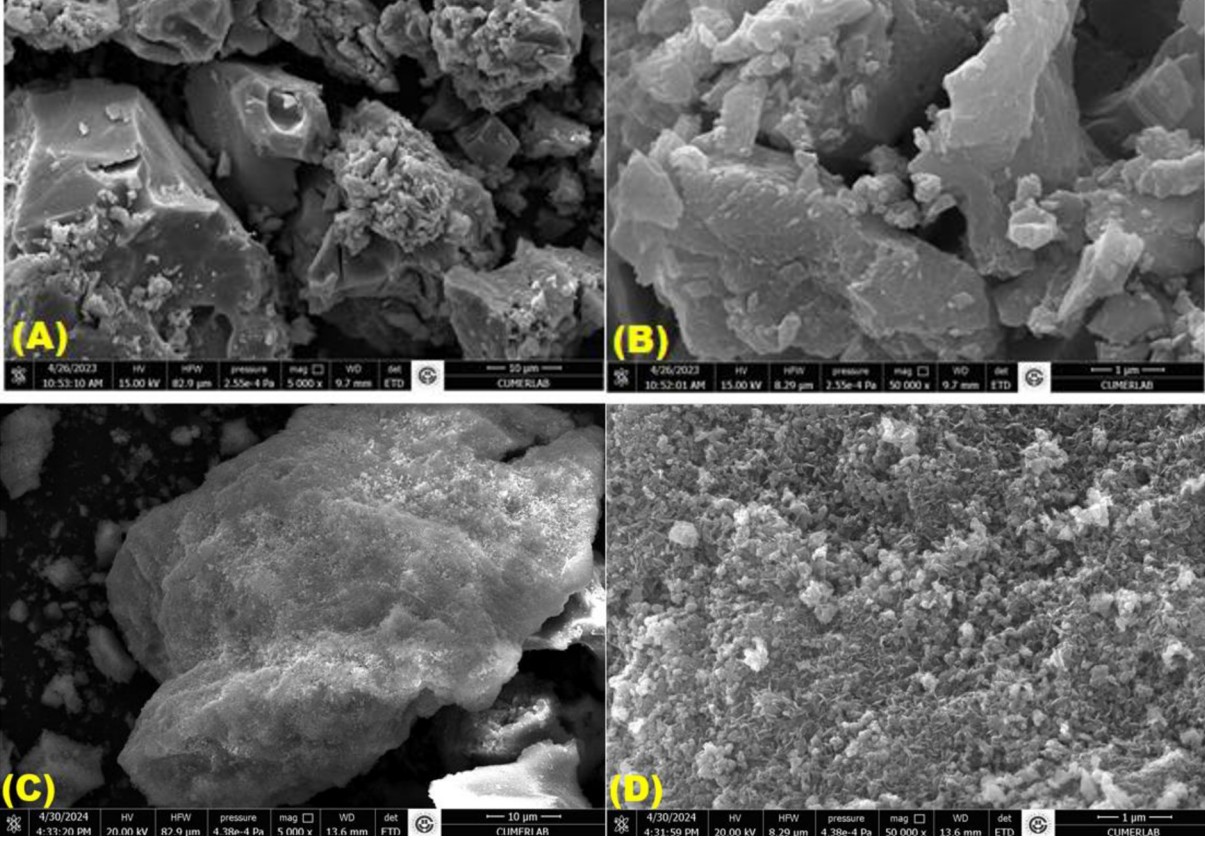

**Fig 2.** **(A-B)** Cr(VI) reduction before, and **(C-D)** Cr(VI) reduction after, SEM images of electro-coagulated Fe₃O₄ sludge.

The magnetic hysteresis curves plot to characterize the magnetic properties of electro-coagulated Fe₃O₄ sludge is shown in Fig 4B. Under an external magnetic field, magnetic fields in magnetic materials align parallel to the magnetic field strength. The direction of the magnetic field creates a magnetic dipole moment, i.e. magnetization. In a sufficiently high magnetic

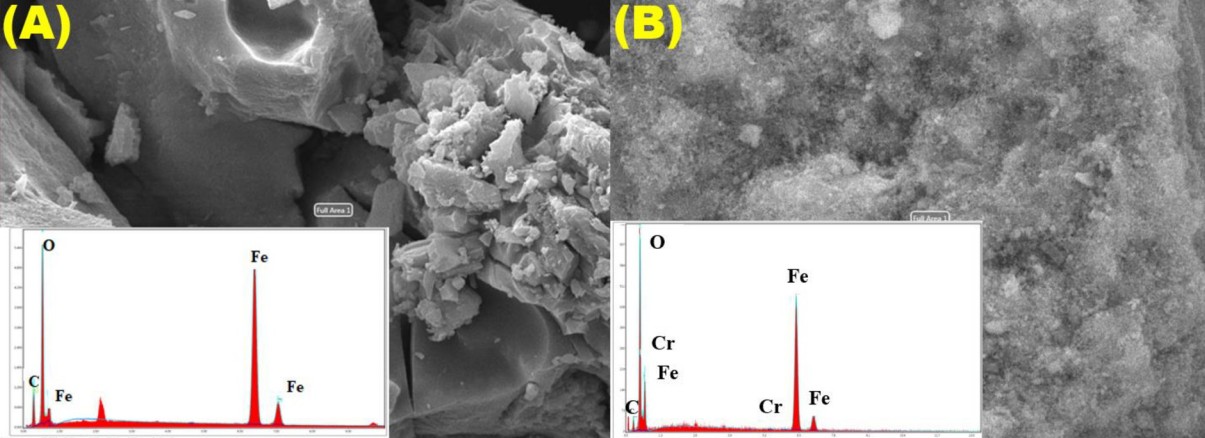

**Fig 3.** **(A)** reduction before and **(B)** after reduction after EDS spectrum of electro-coagulated Fe₃O₄ sludge.

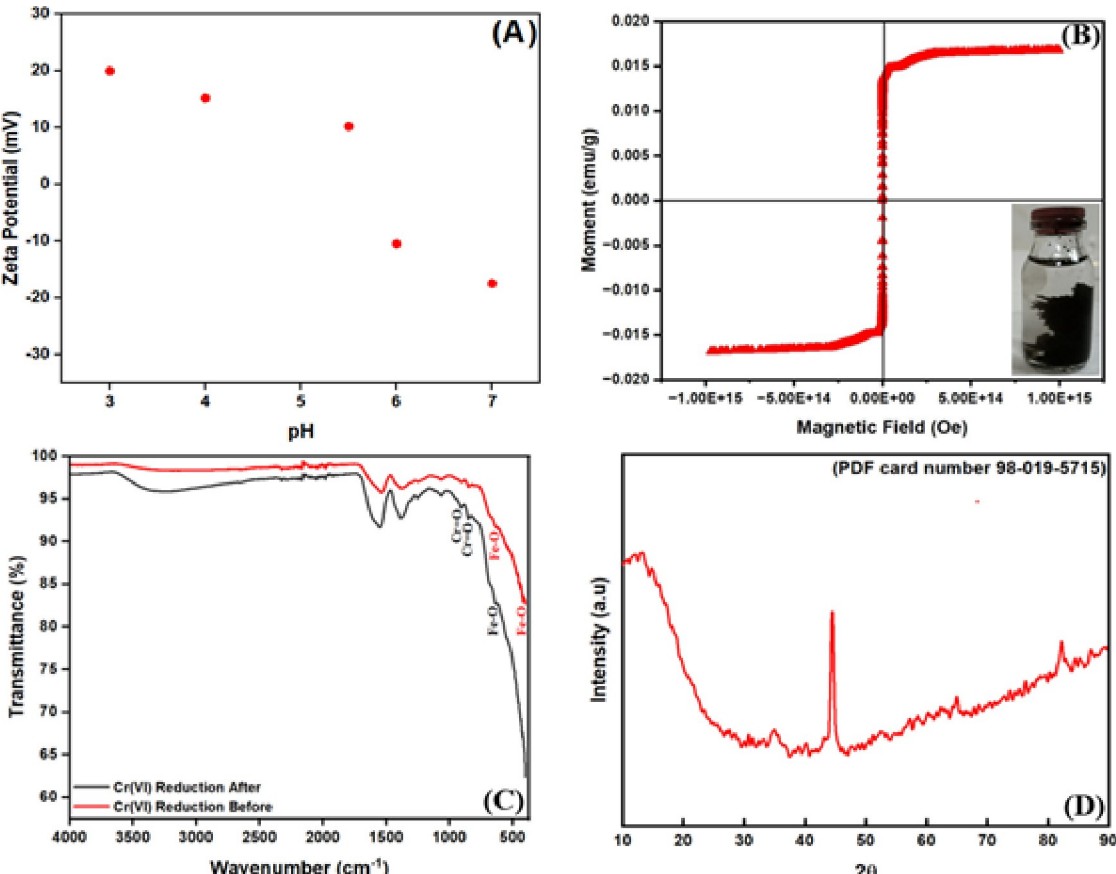

**Fig 4.** (A) Zeta potential, (B) Magnetitc hysteresis curves, (C) FTIR spectrum, and (D) XRD pattern of electro-coagulated $Fe_3O_4$ sludge.

field, magnetization remains almost constant and this value is defined as saturation magnetization (Ms) [33]. In Fig 4B, the saturation magnetization (Ms) of the half-hysteresis loop of ferromagnetic $Fe_3O_4$ nanoparticles was found to be 0.016 emu/g.

The graph of FTIR spectroscopy results for electro-coagulated $Fe_3O_4$ sludge is shown in Fig 4C. The stretching peaks at 628 cm$^{-1}$, 616 cm$^{-1}$, and 420 cm$^{-1}$ belonging to $Fe_3O_4$ in the FTIR spectrum before Cr(VI) reduction are attributed to the presence of $Fe_3O_4$ [34,35]. The stretching peaks at 911 cm$^{-1}$, 881 cm$^{-1}$, and 850 cm$^{-1}$ occurring after Cr(VI) reduction are attributed to the presence of Cr = O [17,18,36]. Post-reduction FTIR spectra showed that functional groups were involved in the Cr(VI) reduction processes.

Upon examining Fig 4D, the XRD results of the electro-coagulated iron-containing sludge reveal diffraction peaks corresponding to $Fe_3O_4$, as indicated by the cubiform structure with card number 98-019-5715 [26]. This analysis confirms the formation of $Fe_3O_4$ structures with magnetic properties in the electro-coagulated iron-containing sludge.

## 3.2. ANOVA statistical results of Cr(VI) reduction using electro-coagulated $Fe_3O_4$ sludge

Table 2 shows the ANOVA statistical analysis results of Cr (VI) reduction, percentage removal efficiency, and adsorption capacity, which were created using the data obtained as a result of the experiments. Whether a model is significant or not depends on its p-value. If the p-value is

**Table 2. ANOVA results of removal efficiency (%) and adsorption capacity (mg/g) of Cr(VI) reduction.**

| Source | Cr(VI) Reduction Efficiency (%) | | | | | Cr(VI) Reduction Adsorption Capacity (mg/g) | | | | |
|---|---|---|---|---|---|---|---|---|---|---|
| | SS | df | MS | F-value | p-value | SS | df | MS | F-value | p-value |
| Model | 18980.35 | 14 | 1355.74 | 105.28 | <0.0001 | 344.02 | 4 | 86.00 | 176.17 | <0.0001 |
| A | 728.26 | 1 | 728.26 | 56.55 | <0.0001 | 42.56 | 1 | 42.56 | 87.19 | <0.0001 |
| B | 2192.74 | 1 | 2192.74 | 170.27 | <0.0001 | 38.88 | 1 | 38.88 | 79.64 | <0.0001 |
| C | 9218.86 | 1 | 9218.86 | 715.87 | <0.0001 | 149.11 | 1 | 149.11 | 305.43 | <0.0001 |
| D | 2100.41 | 1 | 2100.41 | 163.10 | <0.0001 | 113.47 | 1 | 113.47 | 232.42 | <0.0001 |
| AB | 611.58 | 1 | 611.58 | 47.49 | <0.0001 | | | | | |
| AC | 0.0292 | 1 | 0.0292 | 0.0023 | 0.9628 | | | | | |
| AD | 181.96 | 1 | 181.96 | 14.13 | 0.0027 | | | | | |
| BC | 63.37 | 1 | 63.37 | 4.92 | 0.0466 | | | | | |
| BD | 393.21 | 1 | 393.21 | 30.53 | 0.0001 | | | | | |
| CD | 345.84 | 1 | 345.84 | 26.86 | 0.0002 | | | | | |
| $A^2$ | 284.15 | 1 | 284.15 | 22.07 | 0.0005 | | | | | |
| $B^2$ | 65.11 | 1 | 65.11 | 5.06 | 0.0441 | | | | | |
| $C^2$ | 2685.86 | 1 | 2685.86 | 208.56 | <0.0001 | | | | | |
| $D^2$ | 2.52 | 1 | 2.52 | 0.1957 | 0.6661 | | | | | |
| Residual | 154.53 | 12 | 12.88 | | | 10.74 | 22 | 0.4882 | | |
| Lack of Fit | 147.98 | 10 | 14.80 | 4.51 | 0.1949 | 9.91 | 20 | 0.4957 | 1.20 | 0.5511 |
| Pure Error | 6.56 | 2 | 3.28 | | | 0.8267 | 2 | 0.4133 | | |
| Cor Total | 19134.88 | 26 | | | | 354.76 | 26 | | | |

less than 1, it indicates that the model is significant, and if the p-value is greater than 1, it indicates that the model is insignificant [37]. When Table 2 is examined, the models are significant because the p values of the models regarding Cr(VI) removal efficiency and adsorption capacity are less than 1. At the same time, when the Cr (VI) removal efficiency model is examined, the independent variables pH (A), $Fe_3O_4$ sludge dose (B), Cr(VI) concentration (C), reaction time (D), AB, AD, BC, BD, CD, $A^2$, $B^2$ and $C^2$ are significant, while the independent variables AC and $D^2$ are insignificant. When the data on reduction capacity is examined, it is seen that the independent variables A, B, C, and D are important for the model.

For the accuracy of the model, graphs of predicted vs actual and externally studentized residuals of Cr(VI) reduction percentage efficiency and adsorption capacity are given in Fig 5. It confirms that the data obtained according to Fig 5 are closely aligned, highlighting that this is indicative of a good fit without any significant deviations.

Following multiple regression analysis, the resulting Eqs (7) and (8) illustrate the relationship between each response variable and the independent variables, where pH, adsorbent dose, Cr(VI) concentration, and reaction time correspond to A, B, C, and D, respectively.

$$
\begin{aligned}
Cr(VI)&Reduction\ Efficiency(\%) \\
&= +155.90826 - 57.03350A + 199.74715B - 3.39850C + 3.37104D - 24.73022AB \\
&\quad - 0.004272AC - 0.449640AD - 0.398035BC + 1.32197AD - 0.030995CD \\
&\quad + 7.29923A^2 + 13.97580B^2 + 0.056103C^2 - 0.003055D^2
\end{aligned}
\tag{7}
$$

$$
\begin{aligned}
Cr(VI)&Reduction\ Adsorption\ Capacity(mg/g) \\
&= +14.66991 - 1,88333A - 3,60000B + 0,176250C + 0,205000D
\end{aligned}
\tag{8}
$$

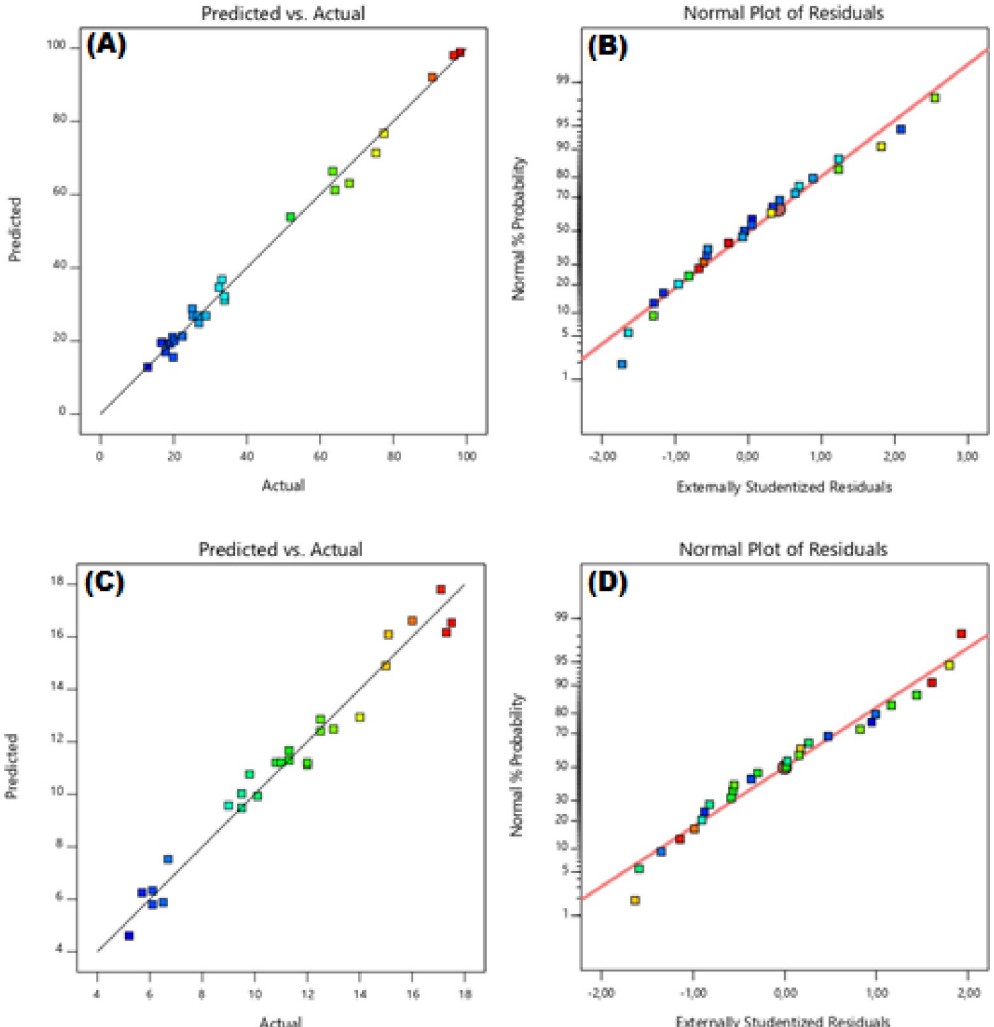

**Fig 5. Predicted vs actual and externally studentized residuals of Cr(VI) reduction.**

### 3.3. The effect of pH on Cr(VI) reduction

The pH level is recognized as one of the most significant parameters that impact Cr reduction. The data illustrating the effect of pH on removal efficiency (%) and adsorption capacity (mg/g) for Cr(VI) reduction by electro-coagulated $Fe_3O_4$ sludge are presented in Fig 6. As depicted, Fig 6 demonstrates a decline in Cr(VI) removal efficiency with increasing pH levels. When it comes to Cr(VI) reduction, the dominant forms of Cr are $HCrO_4$ and $Cr_2O_7-2$, which form between pH 2 and 6 [38]. In this study, it is seen that only the electrostatic attraction force is not responsible for Cr(VI) reduction. In the adsorption process, there are multiple effects rather than a single effect. The most important effect in this study is that electro-coagulated $Fe_3O_4$ sludge plays a role in Cr(VI) reduction by rapidly creating iron ions in its structure under acidic conditions [38]. Iron corrosion usually occurs through the reduction of $H_2O$ when pH is lower than 4.5, whereas when pH is greater than 4.5 the extent of iron corrosion depends on the properties of the oxide films on $Fe^0$. These oxide films depend on factors such as the solubility of iron and the speciation of the contaminant Cr. Therefore, initial low pH values lead to easy dissolution of iron and prevent the immediate formation of oxide films.

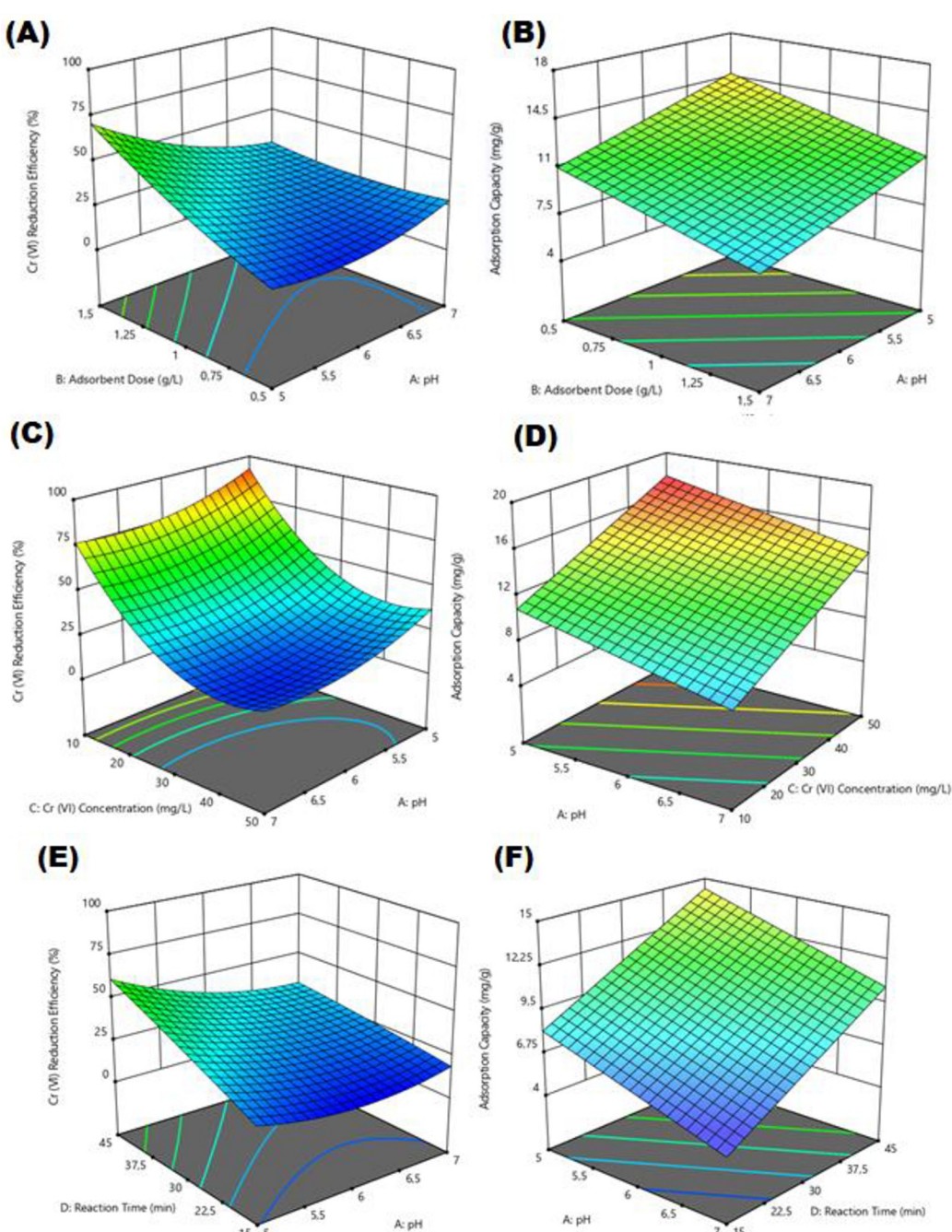

**Fig 6. Impact of pH on removal efficiency (%) and adsorption capacity (mg/g) of Cr(VI) reduction.**

Maintaining a low pH value can prevent oxide film formation and thus increase the reduction rate of Cr(VI). However, the increase in pH value can promote Cr(III) adsorption and co-precipitation [39]. At the same time, according to the zeta potential results, the positive charge on the surface of electro-coagulated $Fe_3O_4$ sludge at pH 5.5. can enhance the reduction of Cr(VI) in the form of $Cr_2O_7^{2-}$ or $HCrO_4^-$ (Fig 4A). Hou et al. (2016) also carried out Cr(VI) removal under acidic conditions and explained the removal mechanism by surrounding the surface of the graphene/$\gamma$-$Fe_2O_3$ adsorbent with $H^+$ ions under acidic conditions and encouraging the

approach of negatively charged $HCrO_4$ through electrostatic attraction [56] (Hou et al., 2016). When the graphs of other parameters such as adsorbent dose, time and chromium concentration are examined in this study, it is seen that the maximum Cr(VI) removal efficiency occurs at pH 5. Therefore, the optimum pH 5 was chosen. In their study, Song et al. [40] used $Fe_3O_4$ nanoribbon@carbon composites for Cr(VI) removal and achieved the maximum removal efficiency at pH 5. They emphasized that the ability of Cr(VI) to significantly increase the adsorption capacity may be due to the synergistic effect between the carbon matrix and Cr(VI), which is increased by the addition of $Fe_3O_4$ [40].

### 3.4. The impact of adsorbent dose on Cr(VI) reduction

Graphs regarding the effect of the amount of electro-coagulated $Fe_3O_4$ sludge on the removal efficiency (%) and adsorption capacity (mg/g) for Cr(VI) reduction are shown in Fig 7. The effect of adsorbent amount on Cr (VI) reduction was examined using the amount of electro-coagulated $Fe_3O_4$ sludge of 0.5–1.5 g/L and the Cr (VI) concentration varying between 10–50 mg/L. Fig 7 shows that as the adsorbent dose increased, the Cr(VI) reduction efficiency increased, but the adsorbent capacity decreased. The increase in Cr (VI) removal efficiency

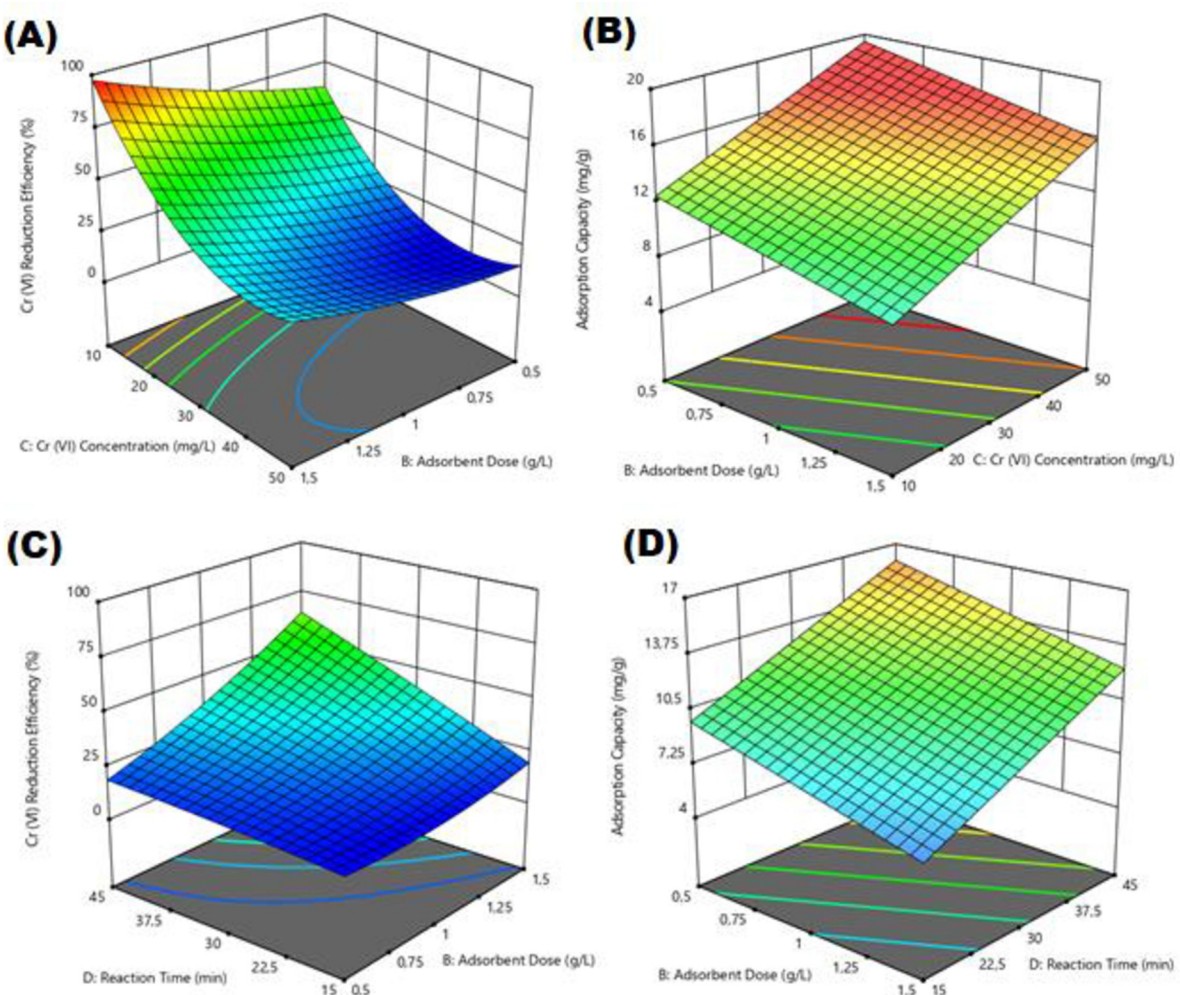

**Fig 7. Impact of Fe₃O₄ sludge dose on removal efficiency (%) and adsorption capacity (mg/g) of Cr(VI) reduction.**

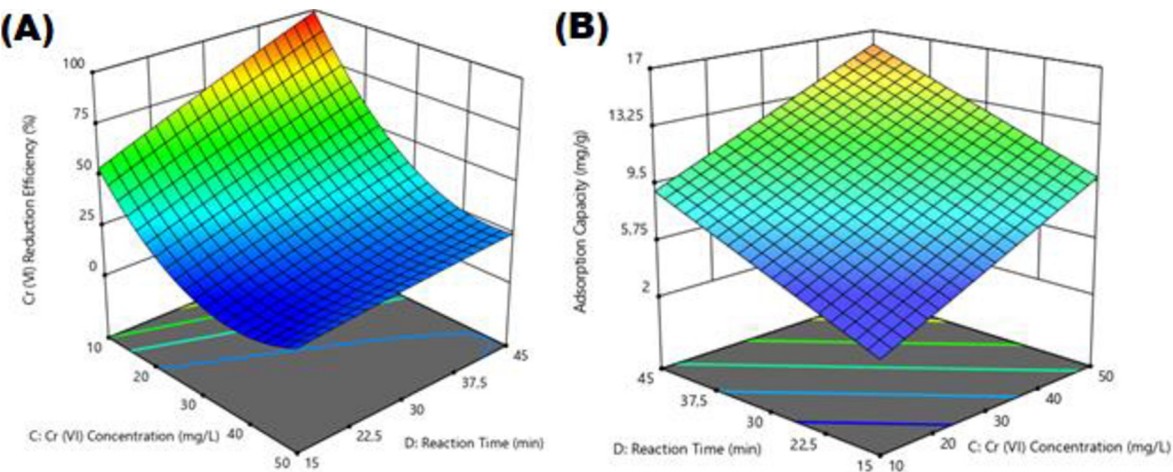

**Fig 8. Effect of Cr(VI) concentration on removal efficiency (%) and adsorption capacity (mg/g) of Cr(VI) reduction.**

with increasing amount of adsorbent is due to the more active sites and functional groups (surface functional groups like Fe–O, Fe–OOH, C double bond O, and O–H) present in the electro-coagulated Fe3O4 sludge [41]. In addition, it should be remembered that increased $Fe_3O_4$ amount may not always lead to an increase in adsorption efficiency, as increased loading may result in a reduction of available adsorption sites and specific surface area [42–44]. In this study, the optimum amount of electro-coagulated Fe3O4 sludge was selected as 1.5 g/L.

### 3.5. The impact of Cr(VI) concentration on Cr(VI) reduction

Graphs for the effect of Cr(VI) concentration on the removal efficiency (%) and adsorption capacity (mg/g) for Cr(VI) reduction are shown in Fig 8. The effect of initial Cr(VI) concentration on Cr (VI) reduction was examined using the amount of electro-coagulated Fe3O4 sludge of 0.5–1.5 g/L, pH of 5.00–7.00, and the Cr (VI) concentration varying between 10–50 mg/L. According to Fig 8, the removal efficiency (%) decreased with increasing initial Cr(VI) concentration. The Cr(VI) removal percentage is initially higher due to the larger surface area of electro-coagulated $Fe_3O_4$ sludge available for Cr(VI) reduction. Once the saturation point is attained, the capacity of the electro-coagulated $Fe_3O_4$ sludge becomes depleted, and the uptake rate becomes regulated by the sorbate transported from the exterior to the interior of the adsorbent particles [45]. The optimum initial Cr(VI) concentration was selected as 10 mg/L.

### 3.6. Possible mechanism of Cr(VI) reduction using electro-coagulated $Fe_3O_4$ sludge

Fig 9 shows the possible reduction mechanism for Cr(VI) reduction using electro-coagulated $Fe_3O_4$ sludge. More than one reduction mechanism is involved in this study. The potential reduction mechanisms can be elucidated as follows: Under acidic conditions, the iron-containing adsorbent swiftly generates iron ions, thereby facilitating Cr(VI) reduction. Simultaneously, the surface of the adsorbent becomes enveloped by $H^+$ ions, promoting the reduction of Cr(VI) ions onto the surface [46,47].

### 3.7. Isotherm and kinetic models

The data obtained as a result of reduction experiments and the parameters of three different isotherm models are calculated from the slope and intersection of the linear graphs. The

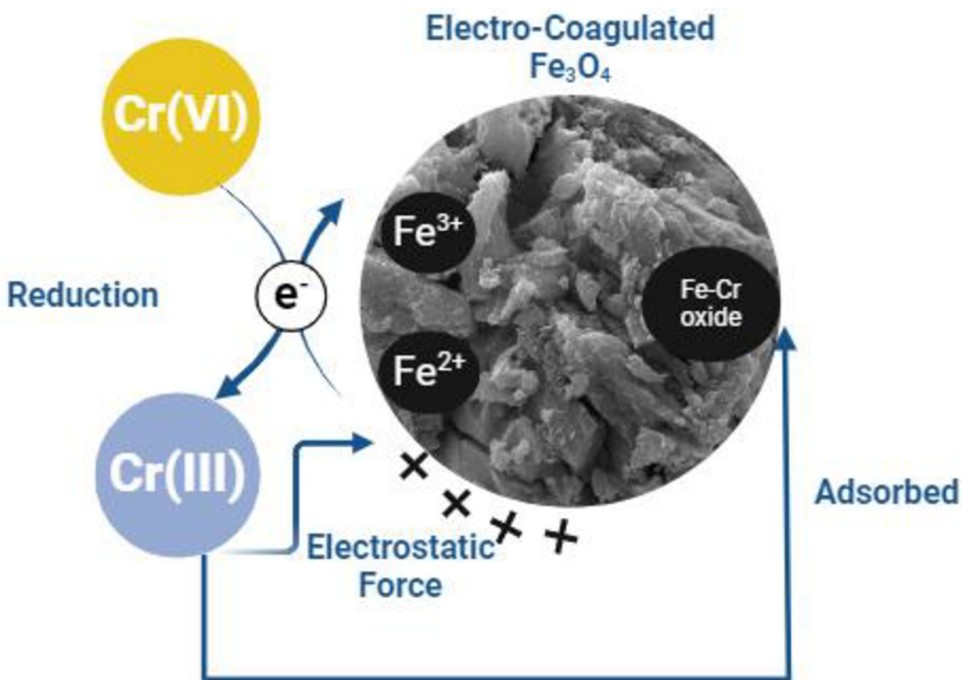

**Fig 9. Possible mechanism of Cr(VI) reduction by $Fe_3O_4$.**

regression coefficients for Langmuir, Freunlich, and D-R are 0.9611, 0.9848, and 0.8757, respectively (Table 3). According to the regression coefficients, the reduction isotherm model is compatible with the Freunlich isotherm model. The n value indicating heterogeneity for the Freunlich isotherm model is more than 1.0 for this study. A value of n greater than 1 indicates that Cr(VI) has more physical reduction. The high value of $K_f$ indicates that there is a close relationship between the adsorbate and the adsorbent [48].

The data obtained as a result of reduction experiments and the parameters of three different kinetic models were calculated from the slope and intersection of linear graphs. Data on regression coefficients and parameters of kinetic models are given in Table 4. According to the regression coefficients of PFO, PSO and ID, they are 0.9918, 0.9959, and 0.9289, respectively. Cr(VI) reduction is compatible with the PFO and PSO kinetic model, and the PSO kinetic model emphasizes that the driving forces of the Cr(VI) reduction process are rapid transfer of Cr(VI) to the adsorbent surface [29].

**Table 3. Isotherm results of Cr(VI) reduction by electro-coagulated $Fe_3O_4$.**

| Isotherm models | Parameter | Data |
| --- | --- | --- |
| **Langmuir** | $K_L$ | 1.46 |
| | $q_{max}$ (mg/g) | 14.72 |
| | $R^2$ | 0.9611 |
| **Freundlich** | $K_f$ (mg/g) | 135.4 |
| | $1/n$ (g/L) | 0.64 |
| | $R^2$ | 0.9848 |
| **D-R** | $K_{ad}$ | 2.62 |
| | $q_s$ (mg/g) | 13.82 |
| | $R^2$ | 0.8757 |

**Table 4. Kinetic models results of Cr(VI) reduction by electro-coagulated $Fe_3O_4$.**

| Kinetics models | Parameters | Data |
|---|---|---|
| **Pseudo-first-order** | $K_1$ ($min^{-1}$) | 0.003 |
| | $q_e$ (mg/g) | 7.29 |
| | $R^2$ | 0.9918 |
| **Pseudo-second-order** | $K_2$ (g/mg.min) | 0.08 |
| | $q_e$ (mg/g) | 3.81 |
| | $R^2$ | 0.9959 |
| **Intraparticle Diffusion** | $K_p$ | 0.36 |
| | C | 2.12 |
| | $R^2$ | 0.9289 |

## 3.8. Thermodynamics models

This analysis aimed to explore the spontaneity, endothermic or exothermic characteristics, and degree of randomness associated with the reduction process [49]. Table 5 presents the parameters derived from Cr(VI) reduction experiments conducted with electro-coagulated $Fe_3O_4$ sludge. Negative $\Delta G°$ values ranging from -4.14 to -5.75 kJ/mol signify the feasibility and spontaneity of the reduction process, with higher temperatures enhancing spontaneity levels. The negative standard enthalpy change ($\Delta H° = -54.61$) indicates an exothermic process, resulting in decreased Cr(VI) reduction at equilibrium with rising temperatures. This phenomenon is governed by weak attractive forces of a physical nature. Moreover, the negative $\Delta S°$ values (-0.16) suggest a tendency towards randomness within the system.

## 3.9. Reuse cycle of electro-coagulated $Fe_3O_4$

The reusability of adsorbents in the reduction process is also a significant factor in evaluating the applicability of the adsorbent. The experiments were carried out without using a desorption agent. To investigate the reusability of electro-coagulated $Fe_3O_4$ sludge, it was carried out under optimum conditions of pH 5.00, electro-coagulated $Fe_3O_4$ amount of 1.5 g/L, 10 mg/L Cr(VI) concentration, and 45 min. Cr(VI) adsorbed electro-coagulated $Fe_3O_4$ sludge was dried and experiments were carried out successively under optimum conditions. Fig 10 shows that the electro-coagulated $Fe_3O_4$ sludge retained 60% of its initial reduction capacity after 5 recyclings. This shows that electro-coagulated $Fe_3O_4$ sludge can be used repeatedly to reduction Cr(VI) from wastewater.

## 3.10. Treatment of real containing Cr(VI) wastewater with electro-coagulated $Fe_3O_4$ sludge

Fig 11 shows the experimental results to determine the optimum catalyst amount of electro-coagulated $Fe_3O_4$ sludge in real wastewater containing Cr(VI). It was determined that the optimum catalyst amount (1.5 g/L) in synthetic wastewater provided 60.0% removal efficiency in real wastewater. Therefore, the amount of electro-coagulated $Fe_3O_4$ sludge was increased to

**Table 5. Thermodynamic results of Cr(VI) reduction by electro-coagulated $Fe_3O_4$.**

| Temperature (K) | $\Delta G$ (kJ.$mol^{-1}$) | $\Delta H$ | $\Delta S$ |
|---|---|---|---|
| **303** | -5.75 | -54.61 | -0.16 |
| **308** | -5.10 | | |
| **313** | -4.14 | | |

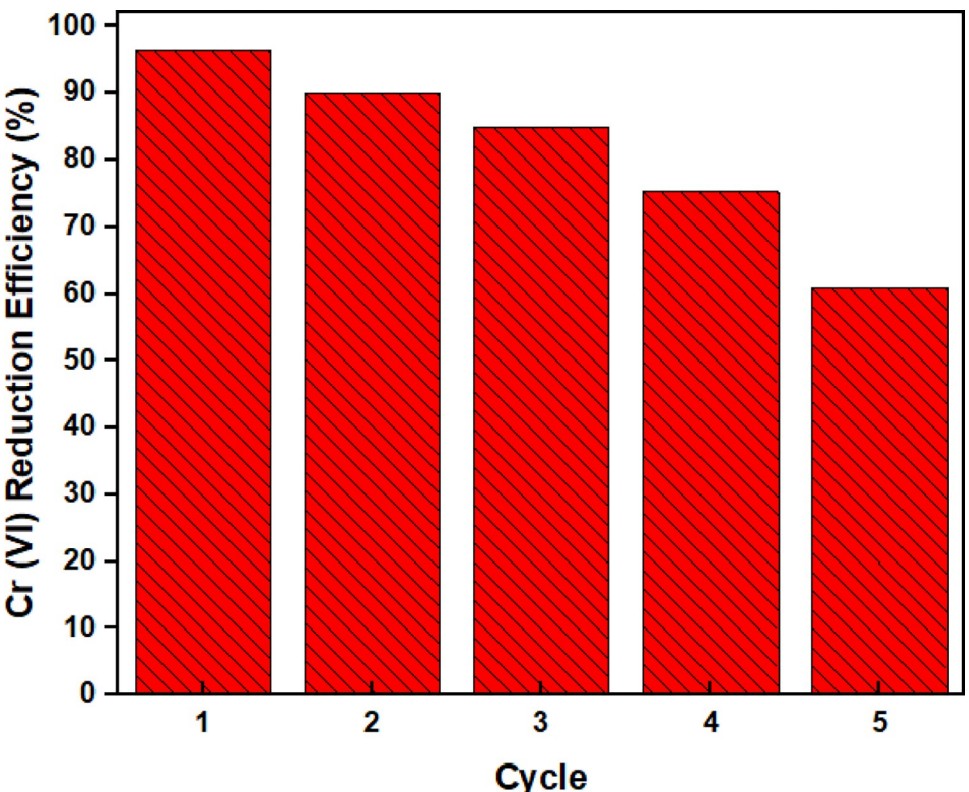

**Fig 10. Reusability of Cr(VI) reduction by electro-coagulated Fe$_3$O$_4$ sludge.**

increase the removal efficiency in real wastewater containing Cr(VI). Removal efficiencies of 60.0%, 62.3%, 70.5%, 75.4% and 77.1% were obtained for catalyst amounts of 1.0 g/L, 1.5 g/L, 2.0 g/L, 2.5 g/L and 5.0 g/L, respectively. By increasing the amount of catalyst up to 5.0 g/L, 75.4% Cr(VI) removal efficiency was obtained for the real wastewater.

### 3.11. Comparison of Cr(VI) reduction with electro-coagulated Fe$_3$O$_4$ sludge with other studies

In this study, the reduction process was employed to assess the applicability of electro-coagulated Fe$_3$O$_4$ sludge. Following optimization studies, the experimental conditions were juxtaposed with those reported in the literature (Table 6). Sharma et al. [50] They investigated the Cr(VI) reduction potential using Fe$_3$O$_4$ and as a result, they obtained 90% Cr(VI) removal efficiency at an initial Cr(VI) concentration of 5 mg/l, pH 6 and 0.5 g/L adsorbent amount [50]. Luo et al. [51] the potential for Cr(VI) removal by using Fe$_3$O$_4$ and MnO$_2$ in multi-walled carbon nanotubes was investigated. They reported 85% removal efficiency at 300 mg/L Cr(VI) concentration and pH 5.00 [51]. Yang et al. [52], synthesized Fe$_3$O$_4$ and Fe$_3$O$_4$@MoS$_2$ particles and investigated the Cr(VI) removal potential. According to the results, 76.07% and 97% removal efficiency of Fe$_3$O$_4$ and Fe$_3$O$_4$@MoS$_2$ particles were obtained, respectively, at pH 5.00 [52]. 99% and 95% Cr(VI) removal efficiency was reported using magnetic iron oxide [53] and Fe$_3$O$_4$-graphene oxide [54] adsorbents, respectively, at pH 2.00. In another study by Soliemanzadeh and Fekri (2017), 90% Cr(VI) removal efficiency was achieved with bentonite-supported zero valent iron adsorbent at pH 5.00 [55]. In this study, Cr(VI) removal efficiency was

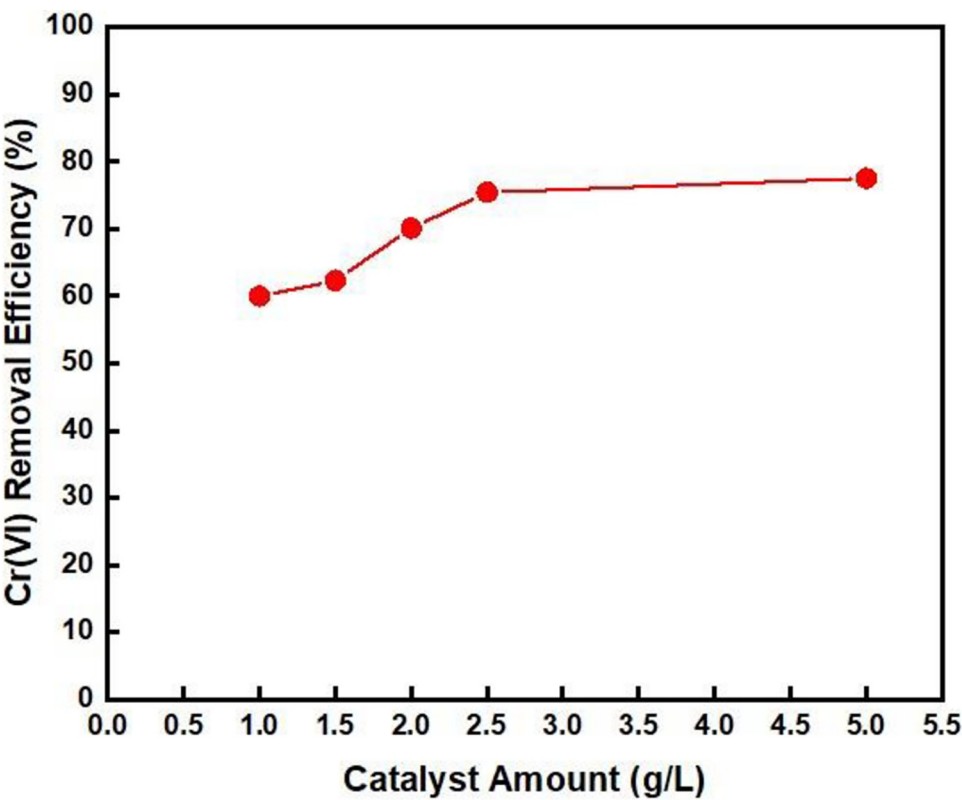

**Fig 11. The amounts of electro-coagulted Fe$_3$O$_4$ sludge used in the treatment of real wastewater containing Cr (VI).**

obtained as 97.00% at pH 5 by using electro-coagulated Fe$_3$O$_4$ sludge. To summarize Table 6, Cr(VI) removal efficiency is quite high at pH 5. Among the adsorbents in the table, the biggest advantage of the electro-coagulated Fe$_3$O$_4$ sludge used in this study is that it is not synthesized chemically like other adsorbents. The two main advantages of this situation are i) that electro-coagulated Fe$_3$O$_4$ sludge is a waste material and reuse of it effectively provides win-win gains, ii) it is not synthesized using chemicals and does not create extra pollution.

**Table 6. Comparison of Cr(VI) reduction with other adsorbents.**

| Adsorbent type | Major finding | Cr (VI) reduction efficiency/capacity | Ref. |
|---|---|---|---|
| **Electro-coagulated Fe$_3$O$_4$ sludge** | 10 mg/L concentration, pH 5.00, Fe$_3$O$_4$ sludge amount 1.5 g/L. | 97.00% | This Study |
| **Fe$_3$O$_4$** | 5 mg/L concentration, pH 6.00, Fe$_3$O$_4$ sludge amount 0.5 g/L. | 90.00% | [50] |
| **MnO$_2$/Fe$_3$O$_4$/o-MWCNTs** | 300 mg/L, pH 5.00. | 85.00% | [51] |
| **Fe$_3$O$_4$ particles** | 20 mg/L concentration, pH 5.00. | 76.07% | [52] |
| **Fe$_3$O$_4$@MoS$_2$ (F@M)** | 20 mg/L concentration, pH 5.00, Adsorbent amount. | 97.00% | [52] |
| **Fe$_3$O$_4$ nanoribbon@ carbon** | 50 mg/L concentration, pH 5.00, 0.25 g/L Fe$_3$O$_4$ sludge amount | 134 mg/g | [56] |
| **Fe$_3$O$_4$-graphene oxide** | 10 mg/L concentration, pH 2.00, 0.6 g/L Fe$_3$O$_4$ sludge amount | 99.00% | [54] |
| **NP-MWCNT composites** | 5 mg/L oncentration, pH 2.00, 1.00 g/L Fe$_3$O$_4$ sludge amount | 95.00% | [53] |
| **Bentonite-supported nanoscale zerovalent Iron** | 100 mg/L concentration, pH 5.00, 2.5 g/L Fe$_3$O$_4$ sludge amount | 90.00% | [55] |

## Conclusion

It was successfully achieved with 97.0% Cr(VI) reduction with electro-coagulated sludge resulting from the electrocoagulation process. The fact that the electro-coagulated sludge has the $Fe_3O_4$ structure adds a unique feature to the adsorbent such that it can be easily separated from the solution. The rapid transformation of iron in the structure of the adsorbent into iron ions in acidic conditions is responsible for Cr(VI) reduction. Compliance with the Freundlich isotherm in the reduction process showed that it had a heterogeneous distribution. Electro-coagulated $Fe_3O_4$ sludge can be reused 5 times to reduce more than 60% of Cr(VI). In light of all these results, electro-coagulated sludge has been shown to be an effective alternative and cheap adsorbent candidate for Cr(VI) reduction from wastewater.

## Supporting information

**S1 Table. Adsorption isotherm for electro-coagulated Fe3O4 sludge.**
(DOCX)

**S2 Table. Adsorption kinetics for electro-coagulated Fe3O4 sludge.**
(DOCX)

## Author Contributions

**Conceptualization:** Pınar Belibagli, Zelal Isik, Nadir Dizge.

**Data curation:** Pınar Belibagli, Zelal Isik.

**Formal analysis:** Pınar Belibagli, Zelal Isik, Nadir Dizge.

**Funding acquisition:** Abdul Rahman Afzal, Muhammad Akram.

**Investigation:** Pınar Belibagli, Zelal Isik, Nadir Dizge.

**Methodology:** Pınar Belibagli, Zelal Isik, Nadir Dizge.

**Project administration:** Deepanraj Balakrishnan, Abdul Rahman Afzal, Muhammad Akram.

**Software:** Deepanraj Balakrishnan, Abdul Rahman Afzal, Muhammad Akram.

**Supervision:** Nadir Dizge, Deepanraj Balakrishnan.

**Validation:** Deepanraj Balakrishnan, Muhammad Akram.

**Writing – original draft:** Pınar Belibagli, Zelal Isik, Nadir Dizge.

**Writing – review & editing:** Deepanraj Balakrishnan, Abdul Rahman Afzal, Muhammad Akram.

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
