## [Decision Letter · Decision Letter 0]

2 Apr 2024

PONE-D-24-08931Optimization of chromium (VI) reduction in aqueous solution using magnetic Fe3O4 sludge resulting from electrocoagulation processPLOS ONE

Dear Dr. Deepanraj,

Thank you for submitting your manuscript to PLOS ONE. After careful consideration, we feel that it has merit but does not fully meet PLOS ONE’s publication criteria as it currently stands. Therefore, we invite you to submit a revised version of the manuscript that addresses the points raised during the review process.

We look forward to receiving your revised manuscript.

Kind regards,

Santhana Krishna Kumar Alagarsamy

Academic Editor

PLOS ONE

Journal Requirements:

4. We note that Figure 1 in your submission contain copyrighted images. All PLOS content is published under the Creative Commons Attribution License (CC BY 4.0), which means that the manuscript, images, and Supporting Information files will be freely available online, and any third party is permitted to access, download, copy, distribute, and use these materials in any way, even commercially, with proper attribution. For more information, see our copyright guidelines: http://journals.plos.org/plosone/s/licenses-and-copyright.

Additional Editor Comments:

I have raised few concerns that might be useful for author’s to improve their manuscript at current stage, after successful implementation the improved manuscript could be considered for further publication.

Editor Comments to the author

Author has prepared Optimization of chromium (VI) reduction in aqueous solution using magnetic Fe3O4 sludge resulting from electrocoagulation process

The experimental studies need to be more systematic and scientifically must be strong enough to support in order to attract this article reader which are lacking in the current form the manuscript, especially

1. In the section 3.3, and Table 2, author mentioned that, Cr(VI) could be reduced to Cr(III), how did you author confirmed their reduction, have you performed high resolution XPS spectra, Electron paramagnetic resonance to confirm their reduction, I suggest the following article might be useful for authors for further understanding to read and refer them Self-Assembly of Poly(ethyleneimine)-Modified g-C3N4 Nanosheets with Lysozyme Fibrils for Chromium Detoxification, Langmuir 2021, 37, 7147−7155,

2. In the section 3 Results and Discussion, Please include FT-IR spectra and also please provide before and after reduction of Cr(VI) based on FT-IR spectra studies this spectra would be helpful to identify their functional groups are Cr-O stretching and Cr=O stretching after sorption of Cr(VI) along with adsorbent for further information please carefully read and refer [Enhanced adsorption of hexavalent chromium arising out of an admirable interaction between a synthetic polymer and an ionic liquid, Chemical Engineering Journal 222 (2013) 454–463 an additionally Ultrasound-assisted preparation and characterization of crystalline cellulose–ionic liquid blend polymeric material: A prelude to the study of its application toward the effective adsorption of chromium, Journal of Colloid and Interface Science 367 (2012) 398–408.

3. In Figure 2 & 3, please discuss SEM images coupled with Energy dispersive spectra analysis (EDS) along with elemental compositions for before and after reduction of Chromium (VI) subsequently discussion need to be incorporate for further, an additional clarification I ask author to go through this research article

An enhanced adsorption methodology for the detoxification of chromium using n-octylamine impregnated Amberlite XAD-4 polymeric sorbent, Journal of Environmental Science and Health, Part A 2011, 46, 1598-1610.

4. In the introduction section, what are the new aspects and originality of this manuscript? authors have to discuss a detailed previously published Chromium (VI) removal and reduction-based literature studies for examples please include

Heavy metal and organic dye removal via a hybrid porous hexagonal boron nitride-based magnetic aerogel, npj Clean Water, 2022, 5, 24. Chemical Engineering Journal 222 (2013) 454–463. Journal of Colloid and Interface Science 367 (2012) 398–408. Langmuir 2021, 37, 7147−7155, rather than discussing (introduction part) about dyes removal and reduction and also how do you correlate from previous studies and what are the additional advantages from existing scientific published articles, a concise literature studies should be added into the introduction part that should be relevant to the prosed adsorbent materials?

5. In the XRD spectra, Figure 4, I didn’t find the existence Fe3+ and Fe2+ ions in the prominent magnetic nanoparticle (Fe3O4) for further information please carefully read this article will be helpful npj Clean Water, 2022, 5, 24.

6. Please provide magnetic hysterias curve for the as synthesised magnetic Fe3O4 sludge nano-adsorbent, provide digital images for suspension solution subsequently showcase after keeping the magnet how this suspension could be resulted as an transparent? For further clarification I ask author to go through this research article for further information’s Heavy metal and organic dye removal via a hybrid porous hexagonal boron nitride-based magnetic aerogel, npj Clean Water, 2022, 5, 24.

Reviewers' comments:

Reviewer's Responses to Questions

**Comments to the Author**

1. Is the manuscript technically sound, and do the data support the conclusions?

Reviewer #1: Yes

Reviewer #2: No

2. Has the statistical analysis been performed appropriately and rigorously? 

Reviewer #1: Yes

Reviewer #2: Yes

3. Have the authors made all data underlying the findings in their manuscript fully available?

Reviewer #1: Yes

Reviewer #2: Yes

4. Is the manuscript presented in an intelligible fashion and written in standard English?

Reviewer #1: Yes

Reviewer #2: No

5. Review Comments to the Author

Reviewer #1: This manuscript reported the “Optimization of chromium (VI) reduction in aqueous solution using magnetic Fe3O4 sludge resulting from electrocoagulation process”. The manuscript needs some alterations before one can take a final decision. This review manuscript is still far from Major Revision in its current status because of following reasons:

1. Check the page for typo errors (e.g., space, superscript, subscript, spelling problems) on individual lines, such as 214, 240, 252, and 332.

2. The author should add a clear figure caption for Figure 5 (A, B, C, D) in the revised manuscript.

3. In section 3.6, the author said that Cr (VI) was reduced to Cr(III) in aqueous solution using a magnetic Fe3O4 sludge catalyst, however the author provided no data or results for the Cr(III) formation. UV-Vis spectra, a solution picture, and XPS data for Cr(III) reduction must all be included in the revised manuscript.

4. In Section 3.9, the author describes how to remove and reuse Cr(VI) from absorbed Fe3O4 sludge.

5. It is possible to test an industrial sample for Cr(VI) reduction using this Fe3O4 catalyst.

6. Line 132: The author said that "zeta potential analyses were performed for the characterization of electro-coagulated sludge." However, no zeta potential data is included in the manuscript. The revised manuscript must include zeta potential data.

7. Additionally, a hysteresis loop spectrum could possibly be provided for magnetic property confirmation.

Reviewer #2: 1. There are many errors in typo, grammatic and sentence formation throughout the manuscript. Authors can revisit the entire manuscript and correct them.

Examples: line 31- Cr(VI) reduction optimization adsorption experimental sets were designed using response surface design (sentence can be reframed).

Line 35- Freundlich, etc.

Line 98- Abbreviate the MH and MOH for better understanding.

Line 138- Abbreviate BBD for better understanding.

Line 214- Fe3O4 change to Fe3O4 , EDX to EDS (throughout the manuscript)

2. Line 69- Sharma et al., 2021 reference is not given in reference section.

3. Line 141- It is better to explain the synthesis procedure of adsorbent for better understanding.

4. Authors used “adsorption”, “reduction”, “removal” of Cr(VI) throughout the manuscript including the title. This may confuse the readers understanding that, what exactly the authors did? Is it reduction of Cr(VI) to Cr(III) using adsorbent? Or removal of Cr(VI) by adsorbent? If the work is focused on reduction of Cr(VI) to Cr(III), it should be mentioned clearly in introduction part.

5. Why authors calculated both adsorption capacity and removal efficiency? How do they differ? They both represent the same fact. Authors can use either adsorption capacity (qe) or removal efficiency (%) throughout the manuscript including figures.

6. Line 172- References can be given for the kinetics models.

7. Line 176- Gibbs free energy ( G), total enthalpy changes ( H), and entropy changes ( S) � is missing.

8. Line 246- Authors can explain more about how increasing iron ions plays a role in Cr(VI) removal.

9. Line 248- “At the same time, H+ in acidic conditions can increase the adsorption of Cr(VI) in the form of Cr2O7 2− or HCrO4 − by neutralizing the negative charge on the surface of electro-coagulated Fe3O4 sludge (Niu et al., 2019).”

How electro-coagulated Fe3O4 sludge surface is negatively charged at acidic pH?

10. Line 272- What are the functional groups present on the surface of electro-coagulated Fe3O4 sludge? Authors can mention them for better understanding.

11. How is adsorption capacity decreasing while increasing the dosage? At the same time removal efficiency increasing with increasing the dosage. Author can explain more about this behaviour.

12. Line 274- How stability of chromium ions affects the adsorption capacity.

13. Line 340- Regeneration method is unclear. How desorption performed? What reagents were used?

14. The deviation of the results were not mentioned for any analysis. Authors should comment on the reproducibility of the results or include error bars.

15. Author mentioned “chemisorption” in line 36 and “physical adsorption” in line 310.

16. Authors should emphasis the novelty of the adsorbent and how this electro-coagulated sludge Fe3O4 is better compared to already reported adsorbents.

6. PLOS authors have the option to publish the peer review history of their article (what does this mean?). If published, this will include your full peer review and any attached files.

Reviewer #1: No

Reviewer #2: No

---

## [Author Response · Author response to Decision Letter 0]

11 Aug 2024

Editor Comments to the author

Author has prepared Optimization of chromium (VI) reduction in aqueous solution using magnetic Fe3O4 sludge resulting from electrocoagulation process. The experimental studies need to be more systematic and scientifically must be strong enough to support in order to attract this article reader which are lacking in the current form the manuscript, especially.

Our Response: We thank the Editor for the comments regarding the presentation of our work. All the comments were corrected.

1. In the section 3.3, and Table 2, author mentioned that, Cr(VI) could be reduced to Cr(III), how did you author confirmed their reduction, have you performed high resolution XPS spectra, Electron paramagnetic resonance to confirm their reduction, I suggest the following article might be useful for authors for further understanding to read and refer them Self-Assembly of Poly(ethyleneimine)-Modified g-C3N4 Nanosheets with Lysozyme Fibrils for Chromium Detoxification, Langmuir 2021, 37, 7147−7155.

Our Response: We thank the editor. Unfortunately, we could not perform XPS and Electron paramagnetic resonance analysis due to lack of the equipment.

2. In the section 3 Results and Discussion, Please include FT-IR spectra and also please provide before and after reduction of Cr(VI) based on FT-IR spectra studies this spectra would be helpful to identify their functional groups are Cr-O stretching and Cr=O stretching after sorption of Cr(VI) along with adsorbent for further information please carefully read and refer [Enhanced adsorption of hexavalent chromium arising out of an admirable interaction between a synthetic polymer and an ionic liquid, Chemical Engineering Journal 222 (2013) 454–463 an additionally Ultrasound-assisted preparation and characterization of crystalline cellulose–ionic liquid blend polymeric material: A prelude to the study of its application toward the effective adsorption of chromium, Journal of Colloid and Interface Science 367 (2012) 398–408.

Our Response: We thank the editor. FTIR results before and after Cr(VI) reduction are attached.

‘The graph of FTIR spectroscopy results for electro-coagulated Fe3O4 sludge is shown in Figure 4C. The stretching peaks at 628 cm-1, 616 cm-1, and 420 cm-1 belonging to Fe3O4 in the FTIR spectrum before Cr(VI) reduction are attributed to the presence of Fe3O4 (Megha et al., 2023; Shi et al., 2023). The stretching peaks at 911 cm-1, 881 cm-1, and 850 cm-1 occurring after Cr(VI) reduction are attributed to the presence of Cr=O (Kalidhasan et al., 2013; Kalidhasan et al., 2012; Holman et al., 1999). Post-reduction FTIR spectra showed that functional groups were involved in the Cr(VI) reduction processes.’

Figure 4. (A) Zeta potential, (B) Magnetitc hysteresis curves, (C) FTIR spectrum, and (D) XRD pattern of electro-coagulated Fe3O4 sludge

3. In Figure 2 & 3, please discuss SEM images coupled with Energy dispersive spectra analysis (EDS) along with elemental compositions for before and after reduction of Chromium (VI) subsequently discussion need to be incorporate for further, an additional clarification I ask author to go through this research article An enhanced adsorption methodology for the detoxification of chromium using n-octylamine impregnated Amberlite XAD-4 polymeric sorbent, Journal of Environmental Science and Health, Part A 2011, 46, 1598-1610.

Our Response: We thank the reviewer Editor. SEM and EDS results after Cr(VI) reduction were added.

‘Figure 2(C-D) shows SEM images after Cr(VI) reduction. When Figure 2(C-D) is examined, it is clearly seen that the structure of the iron-containing electrocoagulation sludge differs after Cr(VI) reduction. Kumar et al., 2011 reported the appearance of shiny particles on the adsorbent surface and stated that there was a change in the structure after Cr(VI) reduction (Kumar et al., 2011).’

Figure 2. (A-B) Cr(VI) reduction before, and (C-D) Cr(VI) reduction after, SEM images of electro-coagulated Fe3O4 sludge

‘Figure 3A shows the EDS spectrum results of Fe3O4. According to EDS spectrum analysis, the dominant elements in the electro-coagulated sludge are 79.65% Fe, 12.85% O and 7.5% C. The 7.5% C element in the structure of electro-coagulated sludge is due to the removal of citric acid (C₆H₈O₇) in electrocoagulation experiments (Belibagli et al., 2024). The graph of EDS after Cr(VI) reduction is shown in Figure 3B. After Cr(VI) reduction, 6.84% C, 30.19% O, 62.57% Fe and 0.40% Cr elements are dominant and these results show that Cr(VI) reduction has occurred.’

Figure 3. (A) reduction before and (B) after reduction after EDS spectrum of electro-coagulated Fe3O4 sludge

4. In the introduction section, what are the new aspects and originality of this manuscript? authors have to discuss a detailed previously published Chromium (VI) removal and reduction-based literature studies for examples please include. Heavy metal and organic dye removal via a hybrid porous hexagonal boron nitride-based magnetic aerogel, npj Clean Water, 2022, 5, 24. Chemical Engineering Journal 222 (2013) 454–463. Journal of Colloid and Interface Science 367 (2012) 398–408. Langmuir 2021, 37, 7147−7155, rather than discussing (introduction part) about dyes removal and reduction and also how do you correlate from previous studies and what are the additional advantages from existing scientific published articles, a concise literature studies should be added into the introduction part that should be relevant to the prosed adsorbent materials?

Our Response: We thank the reviewer #2.

‘Many successful research studies were conducted on Cr(VI) removal using adsorbents with magnetic properties (Tian et al., 2016; Bagbi et al., 2017; Goushki et al., 2022; Feng et al., 2019; Rajput et al., 2016; Wang et al., 2016; Krishna Kumar et al., 2022; Kalidhasan et al., 2013; Kalidhasan et al., 2012; Arputharaj et al., 2021; Sarikhani et al., 2020). However, the magnetic adsorbents used were synthesized by chemical synthesis. Recently, a few studies have eliminated various pollution in water by using aluminum or iron-containing sludges formed as a result of the electrocoagulation process (EC). Electro-coagulation (EC) is a process that produces minimal volumes of sludge primarily composed of metallic oxide/hydroxide.’

5. In the XRD spectra, Figure 4, I didn’t find the existence Fe3+ and Fe2+ ions in the prominent magnetic nanoparticle (Fe3O4) for further information please carefully read this article will be helpful npj Clean Water, 2022, 5, 24.

Our Response: We thank the editor. We disagree with you on this point. XRD results of electro-coagulated Fe3O4 slurry show Fe3O4 structures, but it is amorphous because it is not a controlled synthesis. Because the Fe3O4 sludge produced as a result of electro-coagulation process is used in this study. Therefore, pure crystal structures of Fe3O4 produced chemically should not be expected.

6. Please provide magnetic hysterias curve for the as synthesised magnetic Fe3O4 sludge nano-adsorbent, provide digital images for suspension solution subsequently showcase after keeping the magnet how this suspension could be resulted as an transparent? For further clarification I ask author to go through this research article for further information’s Heavy metal and organic dye removal via a hybrid porous hexagonal boron nitride-based magnetic aerogel, npj Clean Water, 2022, 5, 24.

Our Response: We thank the Editor. The desired digital photograph is shown in Figure 4B.

Reviewer #1:

This manuscript reported the “Optimization of chromium (VI) reduction in aqueous solution using magnetic Fe3O4 sludge resulting from electrocoagulation process”. The manuscript needs some alterations before one can take a final decision. This review manuscript is still far from Major Revision in its current status because of following reasons:

1. Check the page for typo errors (e.g., space, superscript, subscript, spelling problems) on individual lines, such as 214, 240, 252, and 332.,

Our Response: We thank the reviewer #1. All the comments were corrected.

2. The author should add a clear figure caption for Figure 5 (A, B, C, D) in the revised manuscript. 

Our Response: We thank the reviewer #1. The author should add a clear figure caption for Figure 5 (A, B, C, D) in the revised manuscript.

‘Figure 5. Predicted vs actual and externally studentized residuals of Cr(VI) reduction’

3. In section 3.6, the author said that Cr (VI) was reduced to Cr(III) in aqueous solution using a magnetic Fe3O4 sludge catalyst, however the author provided no data or results for the Cr(III) formation. UV-Vis spectra, a solution picture, and XPS data for Cr(III) reduction must all be included in the revised manuscript.

Our Response: We thank the reviewer #1. Unfortunately, we could not perform XPS and Electron paramagnetic resonance analysis due to lack of equipment.

4. In Section 3.9, the author describes how to remove and reuse Cr(VI) from absorbed Fe3O4 sludge.

Our Response: We thank the reviewer #1.

‘The experiments were carried out without using a desorption agent. To investigate the reusability of electro-coagulated Fe3O4 sludge, it was carried out under optimum conditions of pH 5.00, electro-coagulated Fe3O4 amount of 1.5 g/L, 10 mg/L Cr(VI) concentration, and 45 min. Cr(VI) adsorbed electro-coagulated Fe3O4 sludge was dried and experiments were carried out successively under optimum conditions.’

5. It is possible to test an industrial sample for Cr(VI) reduction using this Fe3O4 catalyst.

Our Response: We thank the reviewer #1.

3.10. Treatment of real containing Cr(VI)wastewater with electro-coagulated Fe3O4 sludge

Figure 11 shows the experimental results to determine the optimum catalyst amount of electro-coagulated Fe3O4 sludge in real wastewater containing Cr(VI). It was determined that the optimum catalyst amount (1.5 g/L) determined in synthetic wastewater provided 60.00% removal efficiency in real wastewater. Therefore, the amount of electro-coagulated Fe3O4 sludge was increased to increase the removal efficiency in real wastewater containing Cr(VI). Removal efficiencies of 60.00%, 62.3%, 70.5%, 75.4% and 77.1% were obtained for catalyst amounts of 1.0 g/L, 1.5 g/L, 2.00 g/L, 2.5 g/L and 5.0 g/L, respectively. By increasing the amount of catalyst up to 5.00 g/L, complete Cr(VI) removal efficiency was obtained for the actual wastewater.

Fig 11. The amounts of electro-coagulted Fe3O4 sludge used in the treatment of real wastewater containing Cr(VI)

6. Line 132: The author said that "zeta potential analyses were performed for the characterization of electro-coagulated sludge." However, no zeta potential data is included in the manuscript. The revised manuscript must include zeta potential data.

Our Response: We thank the reviewer #1.

‘In Figure 4A, the surface charge of the electro-coagulated Fe3O4 sludge was measured by zeta potential analysis at different values of pH from 2.00 to 7.00. The surface charge of electrocoagulated Fe3O4 sludge up to pH 3.00 was in the positive region. Beyond pH 3, the surface charges of WWP were in the negative region. When the solution pH increased from 2.00 to 7.00, the surface charge of WWP decreased from +4.24 mV to −30.43 mV.’

7. Additionally, a hysteresis loop spectrum could possibly be provided for magnetic property confirmation.

Our Response: We thank the reviewer #1.

‘The magnetic hysteresis curves plot to characterize the magnetic properties of electro-coagulated Fe3O4 sludge is shown in Figure 4B. Under an external magnetic field, magnetic fields in magnetic materials align parallel to the magnetic field strength. The direction of the magnetic field creates a magnetic dipole moment, i.e. magnetization. In a sufficiently high magnetic field, magnetization remains almost constant and this value is defined as saturation magnetization (Ms) (Jiao et al., 2021). In Figure 4B, the saturation magnetization (Ms) of the half-hysteresis loop of ferromagnetic Fe3O4 nanoparticles was found to be 0.016 emu/g.’

Figure 4B. Magnetic hysteresis curves of electro-coagulated Fe3O4 sludge

Reviewer #2:

1. There are many errors in typo, grammatic and sentence formation throughout the manuscript. Authors can revisit the entire manuscript and correct them.

Examples: line 31- Cr(VI) reduction optimization adsorption experimental sets were designed using response surface design (sentence can be reframed).

Line 35- Freundlich, etc.

Line 98- Abbreviate the MH and MOH for better understanding.

Line 138- Abbreviate BBD for better understanding.

Line 214- Fe3O4 change to Fe3O4, EDX to EDS (throughout the manuscript)

Our Response: We thank the reviewer #2. All the comments were corrected.

2. Line 69- Sharma et al., 2021 reference is not given in reference section.

Our Response: We thank the reviewer #2. Reference added.

Sharma, N., Sodhi, K. K., Kumar, M., & Singh, D. K. (2021). Heavy metal pollution: Insights into chromium eco-toxicity and recent advancement in its remediation. Environmental Nanotechnology, Monitoring & Management, 15, 100388.

3. Line 141- It is better to explain the synthesis procedure of adsorbent for better understanding.

Our Response: We thank the reviewer #2.

‘In this study, the sludge obtained as a result of the optimization of the electrocoagulation process using iron electrodes of citric acid, which was carried out in the previous study (Belibagli et al., 2024), was used. The optimum conditions for the EC process are 20 V, pH 2.0, and a reaction time of 120 min. The sludge used is called electrocoagulated Fe3O4 sludge.’

4. Authors used “adsorption”, “reduction”, “removal” of Cr(VI) throughout the manuscript including the title. This may confuse the readers understanding that, what exactly the authors did? Is it reduction of Cr(VI) to Cr(III) using adsorbent? Or removal of Cr(VI) by adsorbent? If the work is focused on reduction of Cr(VI) to Cr(III), it should be mentioned clearly in introduction part.

Our Response: We thank the reviewer #2. This study is related to Cr(VI) reduction. Therefore, necessary corrections have been made. Adsorption expressions have been changed to reduction.

5. Why authors calculated both adsorption capacity and removal efficiency? How do they differ? They both represent the same fact. Authors can use either adsorption capacity (qe) or removal efficiency (%) throughout the manuscript including figures.

Our Response: We thank the reviewer #2. For a better understanding of the article and in the articles referenced below, both removal efficiency and capacity graphs are given together.

Abubakar, A. M., Schieferstein, E., Zakarya, I. A., Coto, B., Noisri, C., Mobolaji, A. T., & Ahmad, H. (2024). Mathematical models for adsorption capacity and percent removal of heavy metals from water using Stat-Ease 360. Journal of Materials and Engineering (JME), 2(1), 1-19.

Nodeh, M. K. M., Kanani, N., Abadi, E. B., Sereshti, H., Barghi, A., & Rezania, S. (2021). Equilibrium and kinetics studies of naproxen adsorption onto novel magnetic graphene oxide functionalized with hybrid glycidoxy-amino propyl silane. Environmental Challenges, 4, 100106.

Lo, S. F., Wang, S. Y., Tsai, M. J., & Lin, L. D. (2012). Adsorption capacity and removal efficiency of heavy metal ions by Moso and Ma bamboo activated carbons. Chemical Engineering Research and Design, 90(9), 1397-1406.

Gorzin, F., & Bahri Rasht Abadi, M. M. (2018). Adsorption of Cr (VI) from aqueous solution by adsorbent prepared from paper mill sludge: Kinetics and thermodynamics studies. Adsorption Science & Technology, 36(1-2), 149-169.

El Kassimi, A., Achour, Y., El Himri, M., Laamari, R., & El Haddad, M. (2023). Removal of two cationic dyes from aqueous solutions by adsorption onto local clay: experimental and theoretical study using DFT method. International Journal of Environmental Analytical Chemistry, 103(6), 1223-1244.

El-Maghrabi, N., Fawzy, M., & Mahmoud, A. E. D. (2022). Efficient removal of phosphate from wastewater by a novel phyto-graphene composite derived from palm byproducts. ACS omega, 7(49), 45386-45402.

6. Line 172- References can be given for the kinetics models.

Our Response: We thank the reviewer #2. Added references for kinetic models.

Lagergren, S. (1898). About the theory of so-called adsorption of soluble substances.

---

## [Editor Report · Decision Letter 1]

15 Aug 2024

Optimization of chromium (VI) reduction in aqueous solution using magnetic Fe3O4 sludge resulting from electrocoagulation process

PONE-D-24-08931R1

Dear Dr. Muhammad Akram

We’re pleased to inform you that your manuscript has been judged scientifically suitable for publication and will be formally accepted for publication once it meets all outstanding technical requirements.

Kind regards,

Santhana Krishna Kumar Alagarsamy

Academic Editor

PLOS ONE

Additional Editor Comments (optional):

Academic Editor comments

The Academic Editor thanks to the authors for this revised version that address most of the comments made in its previous report. The manuscript can be published as it stands.

---

## [Editor Report · Acceptance letter]

2 Sep 2024

PONE-D-24-08931R1 

PLOS ONE

Dear Dr. Akram, 

I'm pleased to inform you that your manuscript has been deemed suitable for publication in PLOS ONE. Congratulations! Your manuscript is now being handed over to our production team.

Kind regards, 

on behalf of

Dr. Santhana Krishna Kumar Alagarsamy 

Academic Editor

PLOS ONE